# Is Heterophily A Real Nightmare For Graph Neural Networks To Do Node Classification?

## Abstract

Graph Neural Networks (GNNs) extend basic Neural Networks (NNs) by using the graph structures based on the relational inductive bias (homophily assumption). Though GNNs are believed to outperform NNs in real-world tasks, performance advantages of GNNs over graph-agnostic NNs seem not generally satisfactory. Heterophily has been considered as a main cause and numerous works have been put forward to address it. In this paper, we first show that not all cases of heterophily are harmful for GNNs with aggregation operation. Then, we propose new metrics based on a similarity matrix which considers the influence of graph structure and input features on GNNs. The metrics demonstrate advantages over the commonly used homophily metrics by tests on synthetic graphs. From the metrics and the observations, we find some cases of harmful heterophily can be addressed by diversification operation. With this fact and knowledge of filterbanks, we propose the Adaptive Channel Mixing (ACM) framework to adaptively exploit aggregation, diversification and identity operations in each GNN layer to address harmful heterophily. We validate the ACM-augmented baselines with 11 real-world node classification tasks. They consistently achieve significant performance gain and exceed the state-of-the-art GNNs on most of the tasks without incurring significant computational burden.

## 1 Introduction

Deep Neural Networks (NNs) [18] have revolutionized many machine learning areas, including image recognition [17], speech recognition [10] and natural language processing [2], *etc.*One major strength is their capacity and effectiveness of learning latent representation from Euclidean data. Recently, the focus has been put on its applications on non-Euclidean data [4], *e.g.,* relational data or graphs. Combining graph signal processing and convolutional neural networks [19], numerous Graph Neural Networks (GNNs) [29, 7, 12, 30, 15, 24] have been proposed which empirically outperform traditional neural networks on graph-based machine learning tasks, *e.g.,* node classification, graph classification, link prediction and graph generation, *etc.*GNNs are built on the homophily assumption[26], *i.e.,* connected nodes tend to share similar attributes with each other [11], which offers additional information besides node features. Such relational inductive bias [3] is believed to be a key factor leading to GNNs' superior performance over NNs' in many tasks.

Nevertheless, growing evidence shows that GNNs do not always gain advantages over traditional NNs when dealing with relational data. In some cases, even simple Multi-Layer Perceptrons (MLPs) can outperform GNNs by a large margin [35, 22, 5]. An important reason for the performance degradation is believed to be the heterophily problem, *i.e.,* connected nodes tend to have different labels which makes the homophily assumption fail. Heterophily challenge has received attention recently and there are increasing number of models being put forward to address this problem [35, 22, 5, 34, 33].

**Contributions** In this paper, we first demonstrate that not all heterophilous graphs are harmful for aggregation-based GNNs and the existing metrics of homophily are insufficient to decide whether the aggregation operation will make nodes less distinguishable or not. By constructing a similarity matrix from backpropagation analysis, we derive new metrics to depict how much GNNs are influenced by the graph structure and node features. We show the advantage of our metrics over the existing metrics by comparing the ability of characterizing the performance of two baseline GNNs on synthetic graphs of different levels of homophily. From the similarity matrix, we find that diversification operation is able to address some harmful heterophily cases, and then based on which we propose Adaptive Channel Mixing (ACM) GNN framework. The experiments on the synthetic datasets, real-world datasets and the ablation studies consistently show that baseline GNN augmented by ACM framework is able to obtain significant performance boost on node classification tasks on heterophilous graphs.

The rest of this paper is mainly organized as follows: In section 2, we introduce the notation and the background knowledge. In section 3, we conduct node-wise heterophily analysis, derive new metrics based on a similarity matrix and conduct experiments to show their advantage. In section 4.3, we propose the ACM-GNN framework to adaptively utilize the information from different filterbank channels to address heterophily problem. In section 5, we discuss the related works and clarify the differences to our method. In section 6, we provide empirical evaluations on ACM framework, including ablation study and tests on 11 real-world node classification tasks.

## 2 Preliminaries

We introduce the related notation and background knowledge. We use **bold** fonts for vectors (*e.g.,* $\boldsymbol{v}$). Suppose we have an undirected connected graph $\mathcal{G} = (\mathcal{V}, \mathcal{E}, A)$, where $\mathcal{V}$ is the node set with $|\mathcal{V}| = N$; $\mathcal{E}$ is the edge set without self-loop; $A \in \mathbb{R}^{N \times N}$ is the symmetric adjacency matrix with $A_{i,j} = 1$ iff $e_{ij} \in \mathcal{E}$, otherwise $A_{i,j} = 0$. We use $D$ to denote the diagonal degree matrix of $\mathcal{G}$, *i.e.,* $D_{i,i} = d_i = \sum_j A_{i,j}$ and use $\mathcal{N}_i$ to denote the neighborhood set of node $i$, *i.e.,* $\mathcal{N}_i = \{j : e_{ij} \in \mathcal{E}\}$. A graph signal is a vector $\boldsymbol{x} \in \mathbb{R}^N$ defined on $\mathcal{V}$, where $\boldsymbol{x}_i$ is defined on the node $i$. We also have a feature matrix $X \in \mathbb{R}^{N \times F}$, whose columns are graph signals and whose $i$-th row $X_{i,:}$ is a feature vector of node $i$. We use $Z \in \mathbb{R}^{N \times C}$ to denote the label encoding matrix, whose $i$-th row $Z_{i,:}$ is the one-hot encoding of the label of node $i$. The $i$-th column of the identity matrix $I$ is denoted by $\boldsymbol{e}_i$.

### 2.1 Graph Laplacian, Affinity Matrix and Their Variants

The (combinatorial) graph Laplacian is defined as $L = D - A$, which is Symmetric Positive Semi-Definite (SPSD) [6]. Its eigendecomposition gives $L = U\Lambda U^T$, where the columns $\boldsymbol{u}_i$ of $U \in \mathbb{R}^{N \times N}$ are orthonormal eigenvectors, namely the *graph Fourier basis*, $\Lambda = \text{diag}(\lambda_1, \ldots, \lambda_N)$ with $\lambda_1 \leq \cdots \leq \lambda_N$, and these eigenvalues are also called *frequencies*. The graph Fourier transform of the graph signal $\boldsymbol{x}$ is defined as $\boldsymbol{x}_{\mathcal{F}} = U^{-1}\boldsymbol{x} = U^T\boldsymbol{x} = [\boldsymbol{u}_1^T\boldsymbol{x}, \ldots, \boldsymbol{u}_N^T\boldsymbol{x}]^T$, where $\boldsymbol{u}_i^T\boldsymbol{x}$ is the component of $\boldsymbol{x}$ in the direction of $\boldsymbol{u}_i$.

In additional to $L$, some variants are also commonly used, *e.g.,* the symmetric normalized Laplacian $L_{\text{sym}} = D^{-1/2}LD^{-1/2} = I - D^{-1/2}AD^{-1/2}$ and the random walk normalized Laplacian $L_{\text{rw}} = D^{-1}L = I - D^{-1}A$. The affinity (transition) matrices can be derived from the Laplacians, *e.g.,* $A_{\text{rw}} = I - L_{\text{rw}} = D^{-1}A$, $A_{\text{sym}} = I - L_{\text{sym}} = D^{-1/2}AD^{-1/2}$ and are considered to be low-pass filters [25]. Their eigenvalues satisfy $\lambda_i(A_{\text{rw}}) = \lambda_i(A_{\text{sym}}) = 1 - \lambda_i(L_{\text{sym}}) = 1 - \lambda_i(L_{\text{rw}}) \in (-1, 1]$. Applying the renormalization trick [15] to affinity and Laplacian matrices respectively leads to $\hat{A}_{\text{sym}} = \tilde{D}^{-1/2}\tilde{A}\tilde{D}^{-1/2}$ and $\hat{L}_{\text{sym}} = I - \hat{A}_{\text{sym}}$, where $\tilde{A} \equiv A + I$ and $\tilde{D} \equiv D + I$. The renormalized affinity matrix essentially adds a self-loop to each node in the graph, and is widely used in Graph Convolutional Network (GCN) [15] as follows,

$$Y = \text{softmax}(\hat{A}_{\text{sym}} \text{ ReLU}(\hat{A}_{\text{sym}}XW_0) W_1) \tag{1}$$

where $W_0 \in \mathbb{R}^{F \times F_1}$ and $W_1 \in \mathbb{R}^{F_1 \times O}$ are learnable parameter matrices. GCN can be trained by minimizing the following cross entropy loss

$$\mathcal{L} = -\text{trace}(Z^T \log Y) \tag{2}$$

where $\log(\cdot)$ is a component-wise logarithm operation. The random walk renormalized matrix $\hat{A}_{\text{rw}} = \tilde{D}^{-1}\tilde{A}$, which shares the same eigenvalues as $\hat{A}_{\text{sym}}$, can also be applied in GCN. The

85   corresponding Laplacian is defined as $\hat{L}_{\text{rw}} = I - \hat{A}_{\text{rw}}$. $\hat{A}_{\text{rw}}$ is essentially a random walk matrix and
86   behaves as a mean aggregator that is applied in spatial-based GNNs [12, 11]. To bridge the spectral
87   and spatial methods, we use $\hat{A}_{rw}$ in the paper.

## 2.2   Metrics of Homophily

89   The metrics of homophily are defined by considering different relations between node labels and
90   graph structures defined by adjacency matrix. There are three commonly used homophilies: edge
91   homophily [1, 35], node homophily [28], and class homophily [21] [1] defined as follows:

$$H_{\text{edge}}(\mathcal{G}) = \frac{\left|\{e_{uv} \mid e_{uv} \in \mathcal{E}, Z_{u,:} = Z_{v,:}\}\right|}{|\mathcal{E}|}, \quad H_{\text{node}}(\mathcal{G}) = \frac{1}{|\mathcal{V}|} \sum_{v \in \mathcal{V}} \frac{\left|\{u \mid u \in \mathcal{N}_v, Z_{u,:} = Z_{v,:}\}\right|}{d_v},$$

$$H_{\text{class}}(\mathcal{G}) = \frac{1}{C-1} \sum_{k=1}^{C} \left[ h_k - \frac{|\{v \mid Z_{v,k} = 1\}|}{N} \right]_+, \quad h_k = \frac{\sum_{v \in \mathcal{V}} |\{u \mid Z_{v,k} = 1, u \in \mathcal{N}_v, Z_{u,:} = Z_{v,:}\}|}{\sum_{v \in \{v \mid Z_{v,k} = 1\}} d_v}$$

$$(3)$$

92   where $[a]_+ = \max(a, 0)$; $h_k$ is the class-wise homophily metric [21]. They are all in the range
93   of $[0, 1]$ and a value close to 1 corresponds to strong homophily while a value close to 0 indicates
94   strong heterophily. $H_{\text{edge}}(\mathcal{G})$ measures the proportion of edges that connect two nodes in the same
95   class; $H_{\text{node}}(\mathcal{G})$ evaluates the average proportion of edge-label consistency of all nodes; $H_{\text{class}}(\mathcal{G})$
96   tries to avoid the sensitivity to imbalanced class, which can cause $H_{\text{edge}}$ misleadingly large. The
97   above definitions are all based on the graph-label consistency and imply that the inconsistency will
98   cause harmful effect to GNNs. With this in mind, we will show a counter example to illustrate the
99   insufficiency of the above metrics and propose new metrics.

# 3   Analysis of Heterophily

## 3.1   Motivation and Aggregation Homophily

102   Heterophily is believed to be harmful for
103   message-passing based GNNs [35, 28, 5] be-
104   cause intuitively features of nodes in different
105   classes will be falsely mixed and this will lead
106   nodes indistinguishable [35]. Nevertheless, it
107   is not always the case, *e.g.,* the bipartite graph
108   shown in Figure 1 is highly heterophilous ac-
109   cording to the homophily metrics in (3), but
110   after mean aggregation, the nodes in classes 1
111   and 2 only exchange colors and are still dis-
112   tinguishable. Authors in [5] also point out the
113   insufficiency of $H_{\text{node}}$ by examples to show that
114   different graph typologies with the same $H_{\text{node}}$
115   can carry different label information.

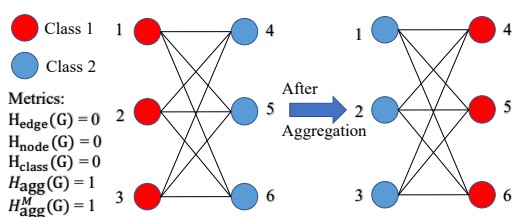

Figure 1: Example of harmless heterophily

116   To analyze to what extent the graph structure can affect the output of a GNN, we first simplify the
117   GCN by removing its non-linearity as [31]. Let $\hat{A} \in \mathbb{R}^{N \times N}$ denote a general aggregation operator.
118   Then, equation (1) can be simplified as,

$$Y = \text{softmax}(\hat{A}XW) = \text{softmax}(Y') \tag{4}$$

119   After each gradient decent step $\Delta W = \gamma \frac{\partial \mathcal{L}}{\partial W}$, where $\gamma$ is the learning rate, and the update of $Y'$ will
120   be (see Appendix B for derivation),

$$\Delta Y' = \hat{A}X\Delta W = \gamma \hat{A}X \frac{\partial \mathcal{L}}{\partial W} \propto \hat{A}X \frac{\partial \mathcal{L}}{\partial W} = \hat{A}XX^T\hat{A}^T(Z - Y) = S(\hat{A}, X)(Z - Y) \tag{5}$$

121   where $S(\hat{A}, X) \equiv \hat{A}X(\hat{A}X)^T$ is a node similarity matrix after aggregation, $Z - Y$ is the prediction
122   error matrix. The update direction of node $i$ is essentially a weighted sum of the prediction error, *i.e.,*
123   $\Delta(Y')_{i,:} = \sum_{j \in \mathcal{V}} \left[ S(\hat{A}, X) \right]_{i,j} (Z - Y)_{j,:}$.

---

[1]The authors in [21] did not name this homophily metric. We name it class homophily based on its definition.

124     To study the effect of heterophily, we define the *aggregation similarity score*.

125     **Definition 1.** *Aggregation similarity score*

$$S_{agg}\left(S(\hat{A}, X)\right) = \frac{\left|\left\{v \,\middle|\, \text{Mean}_u\big(\{S(\hat{A}, X)_{v,u}|Z_{u,:} = Z_{v,:}\}\big) \geq \text{Mean}_u\big(\{S(\hat{A}, X)_{v,u}|Z_{u,:} \neq Z_{v,:}\}\big)\right\}\right|}{|\mathcal{V}|}$$
(6)

126     *where* $\text{Mean}_u\left(\{\cdot\}\right)$ *takes the average over* u *of a given multiset of values or variables.*

127     $S_{\text{agg}}(S(\hat{A}, X))$ is the proportion of nodes $v \in \mathcal{V}$ that will put relatively larger similarity weights on
128     nodes in the same class than in other classes after aggregation. It is easy to see that $S_{\text{agg}}(S(\hat{A}, X)) \in$
129     $[0, 1]$. But in practice, we observe that in most datasets, we will have $S_{\text{agg}}(S(\hat{A}, X)) \geq 0.5$. Based on
130     this observation, we rescale (6) to the following modified aggregation similarity for practical usage,

$$S_{\text{agg}}^M\left(S(\hat{A}, X)\right) = \left[2S_{\text{agg}}\left(S(\hat{A}, X)\right) - 1\right]_+$$
(7)

131     In order to measure the consistency between labels and graph structures without considering node
132     features and make a fair comparison with the existing homophily metrics in (3), we define the graph
133     $(\mathcal{G})$ aggregation $(\hat{A})$ homophily and its modified version as

$$H_{\text{agg}}(\mathcal{G}) = S_{\text{agg}}\left(S(\hat{A}, Z)\right), \ H_{\text{agg}}^M(\mathcal{G}) = S_{\text{agg}}^M\left(S(\hat{A}, Z)\right)$$
(8)

134     In practice, we will only check $H_{\text{agg}}(\mathcal{G})$ when $H_{\text{agg}}^M(\mathcal{G}) = 0$. As Figure 1 shows, when $\hat{A} = \hat{A}_{\text{rw}}$,
135     $H_{\text{agg}}(\mathcal{G}) = H_{\text{agg}}^M(\mathcal{G}) = 1$. Thus, this new metric reflects the fact that nodes in classes 1 and 2 are still
136     highly distinguishable after aggregation, while other metrics mentioned before fail to capture the
137     information and misleadingly give value 0. This shows the advantage of $H_{\text{agg}}(\mathcal{G})$ and $H_{\text{agg}}^M(\mathcal{G})$ by
138     additionally considering information from aggregation operator $\hat{A}$ and the similarity matrix.

139     To comprehensively compare $H_{\text{agg}}^M(\mathcal{G})$ and the metrics in (3) in terms of how they reveal the influence
140     of graph structure on the GNN performance, we generate synthetic graphs and evaluate SGC [31]
141     and GCN [15] on them in the next subsection.

## 3.2    Evaluation and Comparison on Synthetic Graphs

143     **Data Generation & Experimental Setup**     For one dataset, we generate 95 graphs in total with 19
144     edge homophily levels varied from 0.05 to 0.95, each corresponding to 5 graphs. For every generated
145     graph, we have 5 classes with 400 nodes in each class. In each class, there are randomly generated
146     800 intra-class edges and $\left[(800 - 800H_{\text{edge}}(\mathcal{G}))\,/H_{\text{edge}}(\mathcal{G})\right]^2$ inter-class edges. The features of nodes
147     in each class are sampled from node features in the corresponding class of the base dataset. Nodes
148     are randomly split into 60%/20%/20% for train/validation/test. We train 1-hop SGC (*sgc-1* [31] and
149     GCN [15] on synthetic data (see appendix A.1 for hyperparameter searching range). For each value
150     of $H_{\text{edge}}(\mathcal{G})$, we take the average test accuracy and standard deviation of runs over 5 generated graphs.
151     For each generated graph, we also calculate its $H_{\text{node}}(\mathcal{G})$, $H_{\text{class}}(\mathcal{G})$ and $H_{\text{agg}}^M(\mathcal{G})$. Model performance
with respect to different homophily values are shown in Figure 2.

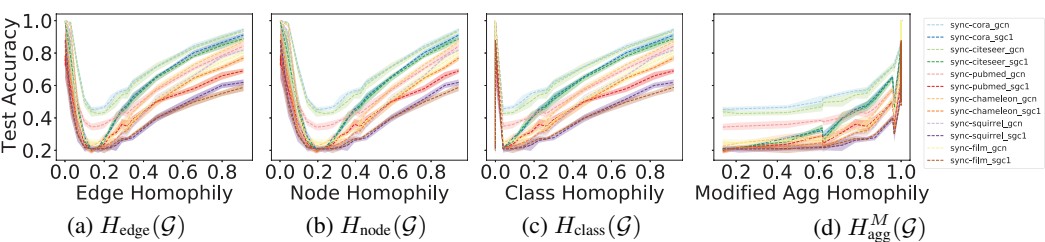

Figure 2: Comparison of baseline performance under different homophily metrics.

152

---

[2]According to (3), $H_{\text{edge}}(\mathcal{G}) = $ #intra-class edges/(#intra-class edges + #inter-class edges)

**Comparison of Homophily Metrics** The performance of SGC-1 and GCN are expected to be monotonically increasing with a proper and informative homophily metric. However, Figure 2(a)(b)(c) show that the performance curves under $H_{\text{edge}}(\mathcal{G})$, $H_{\text{node}}(\mathcal{G})$ and $H_{\text{class}}(\mathcal{G})$ are $U$-shaped [3], while Figure 2(d) reveals a nearly monotonic curve with a little perturbation around 1. This indicates that $H_{\text{agg}}^{M}(\mathcal{G})$ can describe how the graph structure affects the performance of SGC-1 and GCN.

In addition, we notice that in Figure 2(a), both SGC-1 and GCN get the worst performance on all datasets when $H_{\text{edge}}(\mathcal{G})$ is around somewhere between 0.1 and 0.2. This interesting phenomenon can be explained by the following theorem.

**Theorem 1.** (See Appendix C for proof). Suppose there are $C$ classes in the graph $\mathcal{G}$, edges for each node are *i.i.d.* generated such that each edge of any node has probability $h$ of connecting with nodes in the same class and probability $1 - h$ of connecting with nodes in different classes, and $\mathbb{E}(d_v) = d$ for all nodes. Let the aggregation operator $\hat{A} = \hat{A}_{\text{rw}}$. Then, for nodes $v$, $u_1$ and $u_2$, where $Z_{u_1,:} = Z_{v,:}$ and $Z_{u_2,:} \neq Z_{v,:}$, we have

$$g(h) \equiv \mathbb{E}\left(S(\hat{A}, Z)_{v,u_1}\right) - \mathbb{E}\left(S(\hat{A}, Z)_{v,u_2}\right) = \left(\frac{(C-1)(hd+1) - (1-h)d}{(C-1)(d+1)}\right)^2 \quad (9)$$

and the minimum of $g(h)$ is reached at

$$h = \frac{d+1-C}{Cd} = \frac{d_{\text{intra}}/h + 1 - C}{C(d_{\text{intra}}/h)} \Rightarrow h = \frac{d_{\text{intra}}}{Cd_{\text{intra}} + C - 1}$$

where $d_{\text{intra}} = dh$, which is the expectation of the number of neighbors of a node that have the same label as the node.

The value of $g(h)$ in (9) is the expected differences of the similarity values between nodes in the same class as $v$ and nodes in other classes. $g(h)$ is strongly related to the definition of aggregation homophily and its minimum potentially implies the worst value of $H_{\text{agg}}(\mathcal{G})$. In the synthetic experiments, we have $d_{\text{intra}} = 2$, $C = 5$ and the minimum of $g(h)$ is reached at $h = 1/7 \approx 0.14$, which corresponds to the lowest point in the performance curve in Figure 2(a). In other words, the $h$ where SGC-1 and GCN perform worst is where $g(h)$ gets the smallest value, instead of the point with the smallest edge homophily value. This again shows the advantage of $H_{\text{agg}}(\mathcal{G})$ over $H_{\text{edge}}(\mathcal{G})$ by taking use of the similarity matrix.

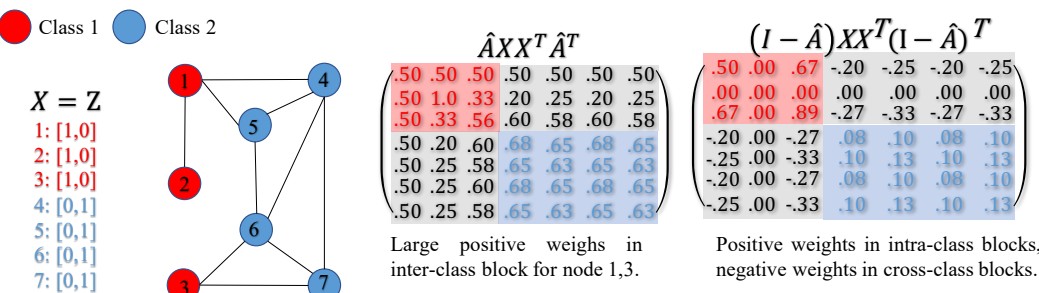

Figure 3: Example of how HP filter addresses harmful heterophily

# 4 Adaptive Channel Mixing (ACM) Framework

## 4.1 How Diversification Operation Helps with Harmful Heterophily

We first consider the example shown in Figure 3. From $S(\hat{A}, X)$, nodes 1,3 assign relatively large positive weights to nodes in class 2, which will negatively affect information aggregation. Despite the fact, we can still distinguish between nodes 1,3 and 4,5,6,7 by considering their neighborhood

---

[3]A similar J-shaped curve is found in [35], though using different data generation processes. It does not mention the insufficiency of edge homophily.

difference: nodes 1,3 are distinguishable from their neighbors while nodes 4,5,6,7 are homogeneous to their neighbors. This indicates, in some cases, although some nodes become similar after aggregation, they are still distinguishable via surrounding dissimilarities. This suggests the possibility of using *diversification operation to address harmful heterophily i.e.,* high-pass (HP) filter $I - \hat{A}$ [8] (will be introduced in next subsection). As $S(I - \hat{A}, Z)$ in Figure 3 shows, nodes 1,3 assign negative weights to nodes 4,5,6,7, *i.e.,* nodes 1,3 treat nodes 4,5,6,7 as negative samples and will move away from them. Base on this example, we propose diversification distinguishability as follows,

**Definition 2.** *Diversification Distinguishability (DD) based on $S(I - \hat{A}, X)$.*

*Given $S(I - \hat{A}, X)$, a node $v$ is diversification distinguishable if the following two conditions are satisfied at the same time,*

$$
\begin{aligned}
&\textbf{1. } \mathrm{Mean}_u \left( \{ S(I - \hat{A}, X)_{v,u} | u \in \mathcal{V} \wedge Z_{u,:} = Z_{v,:} \} \right) \geq 0; \\
&\textbf{2. } \mathrm{Mean}_u \left( \{ S(I - \hat{A}, X)_{v,u} | u \in \mathcal{V} \wedge Z_{u,:} \neq Z_{v,:} \} \right) \leq 0
\end{aligned}
\tag{10}
$$

*Then, graph diversification distinguishability value is defined as*

$$
\mathrm{DD}_{\hat{A},X}(\mathcal{G}) = \frac{1}{|\mathcal{V}|} \Big| \{ v | v \text{ is diversification distinguishable} \} \Big|
\tag{11}
$$

$\mathrm{DD}_{\hat{A},X}(\mathcal{G}) \in [0,1]$ measures the proportion of nodes that HP filter is helpful for. Its effectiveness can be proved for binary classification problems under certain conditions, leading us to:

**Theorem 2.** (See Appendix E for proof). Suppose $X = Z$, $\hat{A} = \hat{A}_{\mathrm{rw}}$. Then, for a binary classification problem, *i.e.,* $C = 2$, all nodes are diversification distinguishable and $\mathrm{DD}_{\hat{A},Z}(\mathcal{G}) = 1$.

Conducting both aggregation and diversification operations to distinctively extract the low- and high-frequency information from graph signals is the same as using filterbanks in graph signal processing. We introduce filterbanks in next subsection.

## 4.2 Filterbank in Spectral and Spatial Forms

**Filterbank**   For the graph signal $x$ defined on $\mathcal{G}$, a 2-channel linear (analysis) filterbank [8] [4] includes a pair of filters $H_{\mathrm{LP}}, H_{\mathrm{HP}}$, where $H_{\mathrm{LP}}$ and $H_{\mathrm{HP}}$ retain the low-frequency and high-frequency content of $x$, respectively.

Most existing GNNs are under uni-channel filtering architecture [15, 30, 12] with either $H_{\mathrm{LP}}$ or $H_{\mathrm{HP}}$ channel that only partially preserves the input information. Unlike the uni-channel architecture, filterbanks with $H_{\mathrm{LP}} + H_{\mathrm{HP}} = I$ will not lose any information of the input signal, *i.e.,* perfect reconstruction property [8, 27].

Generally, the Laplacian matrices ($L_{\mathrm{sym}}, L_{\mathrm{rw}}, \hat{L}_{\mathrm{sym}}, \hat{L}_{\mathrm{rw}}$) can be regarded as HP filters [8] and affinity matrices ($A_{\mathrm{sym}}, A_{\mathrm{rw}}, \hat{A}_{\mathrm{sym}}, \hat{A}_{\mathrm{rw}}$) can be treated as LP filters [25, 11]. Moreover, MLPs can be considered as owing a special identity filterbank with matrix $I$ that satisfies $H_{\mathrm{LP}} + H_{\mathrm{HP}} = I + 0 = I$.

**Filterbank in Spatial Form**   Filterbank methods can also be extended to spatial GNNs. Formally, on the node level, left multiplying $H_{\mathrm{LP}}$ and $H_{\mathrm{HP}}$ on $x$ performs as aggregation and diversification operations, respectively. For example, suppose $H_{\mathrm{LP}} = \hat{A}$ and $H_{\mathrm{HP}} = I - \hat{A}$, then for node $i$ we have

$$
(H_{\mathrm{LP}} x)_i = \sum_{j \in \{ \mathcal{N}_i \cup i \}} \hat{A}_{i,j} x_j, \ (H_{\mathrm{HP}} x)_i = x_i - \sum_{j \in \{ \mathcal{N}_i \cup i \}} \hat{A}_{i,j} x_j
\tag{12}
$$

where $\hat{A}_{i,j}$ is the connection weight between two nodes. To leverage HP and identity channels in GNNs, we propose the Adaptive Channel Mixing (ACM) architecture in the following subsection.

---

[4]In graph signal processing, an additional synthesis filter [8] is required to form the 2-channel filterbank. But synthesis filter is not needed in our framework, so we do not introduce it in our paper.

### 4.3 Adaptive Channel Mixing(ACM) GNN Framework

ACM framework can be applied in lots of baseline GNNs and in this subsection, we use GCN as an example and introduce ACM framework in matrix form. We use $H_{\text{LP}}$ and $H_{\text{HP}}$ to represent general LP and HP filters. The ACM framework includes 3 steps as follows,

**Step 1. Feature Extraction for Each Channel:**

$$H_L^l = H_{\text{LP}}\text{ReLU}\left(H^{l-1}W_L^{l-1}\right),\ H_H^l = H_{\text{HP}}\text{ReLU}\left(H^{l-1}W_H^{l-1}\right),\ H_I^l = I\ \text{ReLU}\left(H^{l-1}W_I^{l-1}\right),$$
$$W_L^{l-1},\ W_H^{l-1},\ W_I^{l-1} \in \mathbb{R}^{F_{l-1}\times F_l};$$

**Step 2. Feature-based Weight Learning with Row Normalization (RN):**

$$\tilde{H}_I^l = \text{RN}\left(H_I^l\right),\ \tilde{H}_L^l = \text{RN}\left(H_L^l\right),\ \tilde{H}_H^l = \text{RN}\left(H_H^l\right);$$
$$\alpha_L^l = \sigma\left(\text{ELU}\left(\tilde{H}_L^l\tilde{W}_L^l\right)\right),\ \alpha_H^l = \sigma\left(\text{ELU}\left(\tilde{H}_H^l\tilde{W}_H^l\right)\right), \alpha_I^l = \sigma\left(\text{ELU}\left(\tilde{H}_I^l\tilde{W}_I^l\right)\right),$$
$$\tilde{W}_L^{l-1},\ \tilde{W}_H^{l-1},\ \tilde{W}_I^{l-1} \in \mathbb{R}^{F_l\times 1};$$

**Step 3. Channel Mixing:**

$$H^l = \left(\text{diag}(\alpha_L^l)H_L^l + \text{diag}(\alpha_H^l)H_H^l + \text{diag}(\alpha_I^l)H_I^l\right).$$

$$(13)$$

ACM-GCN first implements distinct non-linear feature extractions for 3 channels, respectively. After processed by a set of filterbanks, 3 filtered components $H_L^l, H_H^l, H_I^l$ are obtained. Different nodes may have different needs for the information in the 3 channels, *e.g.,* in Figure 3, nodes 1,3 demand high-frequency information while node 2 only needs low-frequency information. To adaptively exploit information from different channels, ACM-GCN learns rowwise (nodewise) feature-conditioned (un-normalized) weights to combine the 3 channels. ACM can be easily plugged into spatial GNNs by replacing $H_{\text{LP}}$ and $H_{\text{HP}}$ by aggregation and diversification operations as shown in (12). See Appendix F for a detailed discussion of model comparison on synthetic datasets.

**Complexity**   Number of learnable parameters in layer $l$ of ACM-GCN is $3F_{l-1}(F_l + 1)$, while it is $F_{l-1}F_l$ in GCN. The computation of step 1-3 takes $NF_l(20 + F_{l-1}) + 2F_l(\text{nnz}(H_{\text{LP}}) + \text{nnz}(H_{\text{HP}}))$ flops, while GCN layer takes $2NF_{l-1}F_l + 2F_l(\text{nnz}(H_{\text{LP}}))$ flops, where $\text{nnz}(\cdot)$ is the number of non-zero elements. A detailed experiments on running time is conducted in section 6.1.

**Limitations**   Diversification operation does not work well in all harmful heterophily cases. For example, consider an imbalanced dataset where several small clusters with distinctive labels are densely connected to a large cluster. In this case, the surrounding differences of nodes in small clusters are similar, *i.e.,* the neighborhood differences are mainly from their connection to the same large cluster, and this possibly makes diversification operation fail to discriminate them. See a more detailed demonstration and discussion in Appendix G.

## 5   Prior Work

**GNNs on Addressing Heterophily**   We discuss relevant work of GNNs on addressing heterophily challenge in this part. [1] acknowledges the difficulty of learning on graphs with weak homophily and propose MixHop to extract features from multi-hop neighborhood to get more information. Geom-GCN [28] precomputes unsupervised node embeddings and uses graph structure defined by geometric relationships in the embedding space to define the bi-level aggregation process. [13] proposes measurements based on feature smoothness and label smoothness that are potentially helpful to guide GNNs on dealing with heterophilous graphs. H$_2$GCN [35] combines 3 key designs to address heterophily: (1) ego- and neighbor-embedding separation; (2) higher-order neighborhoods; (3) combination of intermediate representations. CPGNN [34] models label correlations by the compatibility matrix, which is beneficial for heterophily settings, and propagates a prior belief estimation into GNNs by the compatibility matrix. GPRGNN [5] uses learnable weights that can be both positive and negative for feature propagation, it allows GRPGNN to adapt heterophily structure of graph and is able to handle both high and low frequency parts of the graph signals.

**GNNs with Filterbanks**   Previously, there are geometric scattering networks [9, 27] that apply filterbanks to address over-smoothing [20] problem. The scattering construction captures different channels of variation from node features or labels. In geometric learning and graph signal processing, the band-pass filtering operations extract geometric information beyond smooth signals, thus it is believed that filterbanks can alleviate over-smoothing in GNNs. In ACM framework, we aim to

design a framework with the help of filterbanks to adaptively utilize different channels to address the challenge of learning on heterophilous graph. We deal with different problem as in [9, 27].

# 6 Experiments on Real-World Datasets

In this section, we evaluate ACM framework on real-world datasets. We first conduct ablation studies in subsection 6.1 to validate different components. Then, we compare with the state-of-the-arts models in subsection 6.2.

## 6.1 Ablation Study & Efficiency

| Ablation Study on Different Components in ACM-SGC and ACM-GCN (%) | | | | | | | | | | | | | |
|---|---|---|---|---|---|---|---|---|---|---|---|---|---|
| Baseline | Model Components | | | | Cornell | Wisconsin | Texas | Film | Chameleon | Squirrel | Cora | CiteSeer | PubMed |
| Models | LP | HP | Identity | Mixing | Acc ± Std | Acc ± Std | Acc ± Std | Acc ± Std | Acc ± Std | Acc ± Std | Acc ± Std | Acc ± Std | Acc ± Std |
| SGC-1 w/ | ✓ | | | | 74.43 ± 6.01 | 69.75 ± 5.02 | 84.1 ± 2.32 | 25.34 ± 2.41 | 64.55 ± 1.38 | 42.8 ± 1.1 | 85.24 ± 1.85 | 79.85 ± 1.04 | 84.44 ± 0.38 |
| | ✓ | ✓ | | ✓ | 84.92 ± 4.59 | 91.75 ± 4.05 | 89.34 ± 3.67 | 36.94 ± 1.07 | 63.11 ± 1.64 | 44.8 ± 1.35 | 85.6 ± 1.33 | 80.33 ± 1.25 | 84.5 ± 0.42 |
| | ✓ | | ✓ | ✓ | 92.3 ± 3.8 | 93 ± 2.11 | 91.64 ± 3.65 | 38.25 ± 1.6 | 57 ± 1.93 | 40.2 ± 2.18 | 85.98 ± 0.84 | 80.2 ± 2.01 | 84.37 ± 0.44 |
| | ✓ | ✓ | ✓ | | 88.2 ± 3.88 | 90.75 ± 2.37 | 92.3 ± 3.88 | 36.58 ± 1.36 | 61.64 ± 2.52 | 41.59 ± 2.29 | 84.98 ± 1.2 | 79.81 ± 1.2 | 87.13 ± 0.58 |
| | ✓ | ✓ | ✓ | ✓ | 92.46 ± 2.10 | 93.38 ± 2.68 | 91.97 ± 3.23 | 38.71 ± 1.22 | 62.39 ± 2.45 | 45.65 ± 1.44 | 86.52 ± 1.55 | 80.79 ± 1.65 | 87.69 ± 0.6 |
| GCN w/ | ✓ | | | | 81.31 ± 3.13 | 70.25 ± 4.7 | 82.13 ± 4.05 | 34.45 ± 0.83 | 64.86 ± 1.56 | 45.11 ± 1.39 | 87.47 ± 0.82 | 81.3 ± 0.95 | 87.85 ± 0.44 |
| | ✓ | ✓ | | ✓ | 82.95 ± 5.17 | 88.63 ± 2.51 | 88.03 ± 2.67 | 40.16 ± 1.06 | **68.12 ± 1.73** | 52.08 ± 1.47 | 88.44 ± 1.62 | 81.45 ± 0.9 | 90.09 ± 0.29 |
| | ✓ | | ✓ | ✓ | 92.13 ± 2.65 | 94.37 ± 3.27 | 93.11 ± 2.48 | 40.3 ± 1.63 | 66.67 ± 2.16 | 49.45 ± 0.83 | 88.46 ± 1.31 | 81.42 ± 1.13 | **91.21 ± 1.17** |
| | ✓ | ✓ | ✓ | | 88.52 ± 4.51 | 95 ± 2.25 | 92.3 ± 2.21 | 40.25 ± 1.78 | 65.97 ± 2.24 | 51.02 ± 1.64 | 88.7 ± 1.68 | 80.93 ± 1.53 | 90.66 ± 0.32 |
| | ✓ | ✓ | ✓ | ✓ | **92.62 ± 3.04** | **95.37 ± 2.1** | **94.75 ± 1.77** | **41.48 ± 0.78** | 67.79 ± 1.79 | **52.86 ± 1.96** | **89.11 ± 0.87** | **82.16 ± 0.84** | 90.72 ± 0.7 |
| Average Running Time Per Epoch/Average Total Running Time Comparison | | | | | | | | | | | | | |
| SGC-1 w/ | ✓ | | | | 2.70ms/0.59s | 2.53ms/0.51s | 2.63ms/0.55s | 3.62ms/1.13s | 4.96ms/3.99s | 4.09ms/0.87s | 5.34ms/8.22s | 4.79ms/4.55s | 5.58ms/7.70s |
| | ✓ | ✓ | | ✓ | 4.93ms/1.04s | 5.03ms/1.04s | 6.67ms/1.58s | 6.68ms/1.37s | 6.42ms/1.96s | 7.41ms/1.93s | 6.68ms/2.43s | 6.69ms/1.96s | 7.20ms/2.48s |
| | ✓ | | ✓ | ✓ | 4.73ms/0.98s | 4.99ms/1.09s | 4.79ms/1.02s | 5.53ms/1.28s | 5.89ms/1.50s | 6.48ms/1.50s | 6.50ms/2.09s | 6.23ms/1.76s | 6.73ms/2.24s |
| | ✓ | ✓ | ✓ | | 4.30ms/0.88s | 4.51ms/0.91s | 4.58ms/0.95s | 5.86ms/1.19s | 5.99ms/1.43s | 6.84ms/1.63s | 5.44ms/1.37s | 5.72ms/1.44s | 6.36ms/2.04s |
| | ✓ | ✓ | ✓ | ✓ | 5.15ms/1.08s | 5.82ms/1.28s | 5.55ms/1.18s | 6.28ms/1.50s | 6.60ms/1.96s | 7.27ms/1.52s | 7.05ms/2.40s | 6.99ms/1.94s | 7.28ms/2.07s |
| GCN w/ | ✓ | | | | 3.78ms/0.78s | 3.91ms/0.79s | 3.80ms/0.78s | 4.42ms/0.89s | 4.44ms/0.89s | 6.85ms/1.48s | 4.19ms/0.87s | 5.22ms/1.13s | 4.81ms/0.99s |
| | ✓ | ✓ | | ✓ | 7.63ms/1.54s | 7.99ms/1.92s | 7.26ms/1.48s | 8.42ms/1.73s | 9.74ms/2.76s | 11.19ms/2.38s | 7.74ms/1.61s | 9.98ms/3.56s | 9.10ms/1.85s |
| | ✓ | | ✓ | ✓ | 6.75ms/1.36s | 6.83ms/1.41s | 6.99ms/1.46s | 7.62ms/1.54s | 7.80ms/1.67s | 9.76ms/2.02s | 7.59ms/1.54s | 7.43ms/1.54s | 8.28ms/1.70s |
| | ✓ | ✓ | ✓ | | 7.33ms/1.49s | 6.80ms/1.38s | 6.99ms/1.41s | 8.76ms/2.19s | 7.81ms/1.59s | 11.26ms/2.29s | 7.77ms/1.59s | 7.66ms/1.56s | 8.36ms/1.70s |
| | ✓ | ✓ | ✓ | ✓ | 8.04ms/1.63s | 8.98ms/1.83s | 8.17ms/1.65s | 9.29ms/2.00s | 9.33ms/1.96s | 12.15ms/2.53s | 9.16ms/1.85s | 9.48ms/1.95s | 9.54ms/1.92s |

Table 1: Ablation study on 9 real-world datasets [28]. Cell with ✓ means the component is applied to the baseline model. The best test results are highlighted.

We investigate the effectiveness and efficiency of adding HP, identity channels and the adaptive mixing mechanism in ACM framework by ablation study. Specifically, we apply the above components to SGC-1 and GCN separately, run 10 times on each dataset used in [28] with 60%/20%/20% random splits for train/validation/test and report the average test accuracy as well as the standard deviation. We also record the average running time per epoch(in milliseconds)/average total running time(in seconds) to compare the efficiency. (See Appendix A for hyperparameter searching space.)

From the results we can see that on most datasets, the additional HP and identity channels are helpful, even on strong homophily datasets, such as Cora, CiteSeer and PubMed. The adaptive mixing mechanism also shows its advantage over the method that directly adds the three channels together. This illustrates the necessity of learning to customize the channel usage adaptively for different nodes. As for efficiency, we can see that the running time is approximately doubled in ACM framework than the original model.

## 6.2 Comparison with State-of-the-art Models

**Datasets & Experimental Setup**    In this section, we implement SGC [31] with 1 hop and 2 hop (SGC-1, SGC-2), GCN [15] and GraphSAGE and apply them [12] in ACM framework: we use $\hat{A}_{\mathrm{rw}}$ and mean aggregator as LP filter and the corresponding HP filter can be derived from (12). We compare them with several baselines and state-of-the-art models: MLP with 2 layers (MLP-2), GAT [30], APPNP [16], GPRGNN [5], $H_2$GCN [35], MixHop [1], GCN+JK [15, 32, 21], GAT+JK [30, 32, 21] and Geom-GCN [28]. Besides the 9 benchmark datasets used in [28], we further tests the above models on 2 new benchmark datasets, *Deezer-Europe* and *YelpChi*, that are proposed in

| | Cornell | Wisconsin | Texas | Film | Chameleon | Squirrel | Deezer-Europe | YelpChi | Cora | CiteSeer | PubMed | |
|---|---|---|---|---|---|---|---|---|---|---|---|---|
| #nodes | 183 | 251 | 183 | 7,600 | 2,277 | 5,201 | 28,281 | 45,954 | 2,708 | 3,327 | 19,717 | |
| #edges | 295 | 499 | 309 | 33,544 | 36,101 | 217,073 | 92,752 | 3,846,979 | 5,429 | 4,732 | 44,338 | |
| #features | 1,703 | 1,703 | 1,703 | 931 | 2,325 | 2,089 | 31,241 | 32 | 1,433 | 3,703 | 500 | |
| #classes | 5 | 5 | 5 | 5 | 5 | 5 | 2 | 2 | 7 | 6 | 3 | |
| $H_{edge}(\mathcal{G})$ | 0.5669 | 0.4480 | 0.4106 | 0.3750 | 0.2795 | 0.2416 | 0.5251 | 0.7730 | 0.8100 | 0.7362 | 0.8024 | |
| $H_{node}(\mathcal{G})$ | 0.3855 | 0.1498 | 0.0968 | 0.2210 | 0.2470 | 0.2156 | 0.5299 | 0.7698 | 0.8252 | 0.7175 | 0.7924 | |
| $H_{class}(\mathcal{G})$ | 0.0468 | 0.0941 | 0.0013 | 0.0110 | 0.0620 | 0.0254 | 0.0304 | 0.0520 | 0.7657 | 0.6270 | 0.6641 | |
| $H_{agg}^{M}(\mathcal{G})$ | 0.8032 | 0.7768 | 0.694 | 0.6822 | 0.61 | 0.3566 | 0.5790 | 0.7206 | 0.9904 | 0.9826 | 0.9432 | |
| Data Splits(%) | 60/20/20 | 60/20/20 | 60/20/20 | 60/20/20 | 60/20/20 | 60/20/20 | 50/25/25 | 50/25/25 | 60/20/20 | 60/20/20 | 60/20/20 | |
| | Test Accuracy (%) of State-of-the-art Models, Baseline GNN Models and ACM-GNN models | | | | | | | | | | | Rank |
| MLP-2* | $91.30 \pm 0.70$ | $93.87 \pm 3.33$ | $92.26 \pm 0.71$ | $38.58 \pm 0.25$ | $46.72 \pm 0.46$ | $31.28 \pm 0.27$ | $66.55 \pm 0.72$ | $87.94 \pm 0.52$ | $76.44 \pm 0.30$ | $76.25 \pm 0.28$ | $86.43 \pm 0.13$ | 9.00 |
| GAT* | $76.00 \pm 1.01$ | $71.01 \pm 4.66$ | $78.87 \pm 0.86$ | $35.98 \pm 0.23$ | $63.9 \pm 0.46$ | $42.72 \pm 0.33$ | $61.09 \pm 0.77$ | $81.42 \pm 2.12$ | $76.70 \pm 0.42$ | $67.20 \pm 0.46$ | $83.28 \pm 0.12$ | 11.73 |
| APPNP* | $91.80 \pm 0.63$ | $92.00 \pm 3.59$ | $91.18 \pm 0.70$ | $38.86 \pm 0.24$ | $51.91 \pm 0.56$ | $34.77 \pm 0.34$ | $67.21 \pm 0.56$ | $75.60 \pm 0.48$ | $79.41 \pm 0.38$ | $68.59 \pm 0.30$ | $85.02 \pm 0.09$ | 9.09 |
| GPRGNN* | $91.36 \pm 0.70$ | $93.75 \pm 2.37$ | $92.92 \pm 0.61$ | $39.30 \pm 0.27$ | $67.48 \pm 0.40$ | $49.93 \pm 0.53$ | $66.90 \pm 0.50$ | $71.59 \pm 0.38$ | $79.51 \pm 0.36$ | $67.63 \pm 0.38$ | $85.07 \pm 0.09$ | 6.64 |
| H$_2$GCN | $86.23 \pm 4.71$ | $87.5 \pm 1.77$ | $85.90 \pm 3.53$ | $38.85 \pm 1.17$ | $52.30 \pm 0.48$ | $30.39 \pm 1.22$ | $\textbf{67.22} \pm \textbf{0.90}$ | $88.48 \pm 0.21$ | $87.52 \pm 0.61$ | $79.97 \pm 0.69$ | $87.78 \pm 0.28$ | 7.27 |
| MixHop | $60.33 \pm 28.53$ | $77.25 \pm 7.80$ | $76.39 \pm 7.66$ | $33.13 \pm 2.40$ | $36.28 \pm 10.22$ | $24.55 \pm 2.60$ | $66.80 \pm 0.58$ | $87.02 \pm 0.50$ | $65.65 \pm 11.31$ | $49.52 \pm 13.35$ | $87.04 \pm 4.10$ | 13.09 |
| GCN+JK | $66.56 \pm 13.82$ | $62.50 \pm 15.75$ | $80.66 \pm 1.91$ | $32.72 \pm 2.62$ | $64.68 \pm 2.85$ | $\textbf{53.40} \pm \textbf{1.90}$ | $60.99 \pm 0.14$ | $64.35 \pm 0.86$ | $86.90 \pm 1.51$ | $73.77 \pm 1.85$ | $90.09 \pm 0.68$ | 10.09 |
| GAT+JK | $74.43 \pm 10.24$ | $69.50 \pm 3.12$ | $75.41 \pm 7.18$ | $35.41 \pm 0.97$ | $\textbf{68.14} \pm \textbf{1.18}$ | $52.28 \pm 3.61$ | $59.66 \pm 0.92$ | $\textbf{90.04} \pm \textbf{0.61}$ | $\textbf{89.52} \pm \textbf{0.43}$ | $74.49 \pm 2.76$ | $89.15 \pm 0.87$ | 8.27 |
| Geom-GCN† | 60.81 | 64.12 | 67.57 | 31.63 | 60.9 | 38.14 | NA | NA | 85.27 | 77.99 | 90.05 | 12.33 |
| SGC-1 | $74.43 \pm 6.01$ | $69.75 \pm 5.02$ | $84.1 \pm 2.42$ | $25.34 \pm 3.41$ | $62.34 \pm 1.92$ | $42.8 \pm 1.1$ | $59.73 \pm 0.12$ | $58.62 \pm 0.85$ | $85.16 \pm 0.82$ | $79.93 \pm 1.03$ | $80.97 \pm 0.91$ | 12.18 |
| SGC-2 | $77.7 \pm 4.47$ | $72.75 \pm 3.91$ | $81.48 \pm 3.88$ | $29.39 \pm 0.20$ | $63.02 \pm 0.43$ | $37.41 \pm 1$ | $61.56 \pm 0.51$ | $57.18 \pm 0.75$ | $86.58 \pm 0.26$ | $76.23 \pm 0.29$ | $81.14 \pm 0.71$ | 11.73 |
| GCN | $81.31 \pm 3.13$ | $70.25 \pm 4.7$ | $82.13 \pm 4.05$ | $34.45 \pm 0.83$ | $64.86 \pm 1.56$ | $45.11 \pm 1.39$ | $62.23 \pm 0.53$ | $63.62 \pm 1.00$ | $87.47 \pm 0.82$ | $81.3 \pm 0.95$ | $87.85 \pm 0.44$ | 8.18 |
| GraphSAGE | $71.41 \pm 1.24$ | $64.85 \pm 5.14$ | $79.03 \pm 1.20$ | $36.37 \pm 0.21$ | $62.15 \pm 0.42$ | $41.26 \pm 0.26$ | $62.55 \pm 0.48$ | $62.57 \pm 1.12$ | $86.58 \pm 0.26$ | $78.24 \pm 0.30$ | $86.85 \pm 0.11$ | 11.00 |
| ACM-SGC-1 | $91.31 \pm 2.94$ | $93.38 \pm 2.68$ | $91.97 \pm 3.23$ | $38.71 \pm 1.22$ | $62.39 \pm 2.45$ | $45.65 \pm 1.44$ | $66.42 \pm 0.96$ | $85.83 \pm 1.34$ | $86.52 \pm 1.55$ | $80.79 \pm 1.65$ | $87.69 \pm 0.6$ | 6.27 |
| ACM-SGC-2 | $90.66 \pm 3.36$ | $92.13 \pm 5.06$ | $90.66 \pm 2.84$ | $38.77 \pm 1.74$ | $58.51 \pm 2.42$ | $39.37 \pm 1.41$ | $66.98 \pm 0.88$ | $85.84 \pm 1.17$ | $87.44 \pm 0.8$ | $80.03 \pm 1.26$ | $88.01 \pm 0.93$ | 6.73 |
| ACM-GCN | $\textbf{92.62} \pm \textbf{3.04}$ | $\textbf{95.37} \pm \textbf{2.1}$ | $\textbf{95.08} \pm \textbf{1.8}$ | $\textbf{41.48} \pm \textbf{0.78}$ | $67.79 \pm 1.79$ | $52.86 \pm 1.96$ | $66.85 \pm 0.95$ | $89.91 \pm 1.02$ | $89.11 \pm 0.87$ | $\textbf{82.16} \pm \textbf{0.84}$ | $\textbf{90.72} \pm \textbf{0.7}$ | **1.73** |
| ACM-SAGE | $91.31 \pm 2.94$ | $90.13 \pm 2.67$ | $91.97 \pm 3.15$ | $36.68 \pm 2.46$ | $61.84 \pm 2.71$ | $44.63 \pm 3.02$ | $66.21 \pm 0.89$ | $88.73 \pm 1.45$ | $86.24 \pm 1.25$ | $80.87 \pm 1.36$ | $88.51 \pm 0.9$ | 6.45 |

Table 2: Experimental results: average test accuracy $\pm$ standard deviation on 11 real-world benchmark datasets. The best results are highlighted. The "†" results are from [28] and NA means the reported results are not available. Results "*" are from [5, 21].

[21][5]. We test these models 10 times on *Cornell*, *Wisconsin*, *Texas*, *Film*, *Chameleon*, *Squirrel*, *Cora*, *Citeseer* and *Pubmed* following the same early stopping strategy, the same data splitting and Adam [14] optimizer used in GPRGNN [5]. For *Deezer-Europe* and *YelpChi*, we test the above models 5 times with the same early stopping strategy, the same splits and AdamW [23] used in [21]. The details of hyperparameter search are reported in appendix A.

The main results of this set of experiments with statistics of datasets are summarized in Table 2, where we report the mean accuracy and standard deviation. We can see that after applied in ACM framework, the performance of baseline models are boosted on almost all tasks. Especially, ACM-GCN performs the best in terms of average rank (1.73) across all datasets and achieves SOTA performance on 6 out of 11 datasets. Overall, It suggests that ACM framework can help GNNs to generalize better on node classification tasks on heterophilous graphs.

# 7   Future Work

The similarity matrix and the new metrics defined in this paper mainly capture the linear relations of the aggregated nodes. But this might be insufficient sometimes when nonlinearity information in feature vectors are important for classification. In the future, similarity matrix that is able to capture nonlinear relations between nodes can be proposed to define new homophily metrics.

From experimental results, the standard deviation of ACM-GNNs are relatively higher than GNNs on some tasks and this is suspiciously caused by the feature-based weight learning mechanism. In the future, a stabilizer or a more robust weight learning method can be proposed to reduce the variance.

# 8   Social Impact

We do not find any direct path of this work to any negative social impact.

---

[5]The authors proposed 8 new datasets. From the reported results, GCN only underperform MLP-2 on *Deezer-Europe* and *YelpChi*, which demonstrates the heterophily of these 2 datasets, therefore we choose them.

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
