# OpenReview forum: "Is Heterophily A Real Nightmare For Graph Neural Networks Performing Node Classification?"
_NeurIPS.cc/2021/Conference — NeurIPS 2021 Submitted_

### Official Review · Reviewer_bgkm · 2021-07-12

**Rating:** 6
**Confidence:** 3

**Summary:**

This paper investigates the influence of heterophily on the node classification task of GNNs. They found that not all heterophily cases are baneful for GNNs and propose a similarity matrix to provide a more accurate description of the heterophily degree of graphs. Compared with existing matrices, the proposed matrix can provide more reasonable insights. After, they propose an Adaptive Channel Mixing (ACM) framework to unleash the power of vanilla GNNs on heterophily graphs.


**Limitations And Societal Impact:**

The authors adequately addressed the limitations and potential negative societal impact of their work

**Main Review:**

The paper can be split into several parts: introducing new metric for measuring heterophily, proposing adaptive channel mixing framework, and experimental comparison.

The new metric is nicely explained with a motivational example. It's clear that the value of the new metric is different from the commonly used ones. However, I'm not certain how theorem 1 explains the performance of the models on low-homphily regime and how it supports the aforementioned claim.

The proposed model is based on adaptive channel mixing, which introduces low-pass and high-pass filters. This is very similar to the work [1] in which the model FAGCN combines two signals in the same model. However, this model is not discussed and compared in the experiments. Adding the discuss and comparison with this model is essential for this work.

Experiments thoroughly demonstrate that the proposed ACM-GNN model achieves SOTA performance on many datasets.

Other questions to the authors:
* Why there is a little bump in Figure 2d at the high values of H_agg?

[1] Beyond Low-frequency Information in Graph Convolutional Networks. Bo et al. 2021-AAAI

**Time Spent Reviewing:**

20

---

> ### Author Response · Authors · 2021-08-10
> **Response to Reviewer bgkm**
>
> Thanks for your suggestions and positive comments and here are our responses to your concerns:
>
>
> Q1:"... I'm not certain how theorem 1 explains the performance of the models on low-homphily regime and how it supports the aforementioned claim."
>
>
> R1: It is believed by many people that GNN models perform worse on graphs with lower homophily values.
> Theorem 1 gives a different perspective based on our proposed method that the lowest points of the performance curves does not corresponds to the lowest edge homophily value, i.e. $h=0$. This matches the observation of our synthetic experiments and lead us to put forward the ACM framework.
>
> Q2:" ...This is very similar to the work [1] in which the model FAGCN combines two signals in the same model. However, this model is not discussed and compared in the experiments. Adding the discuss and comparison with this model is essential for this work."
>
> R2: Thanks for kindly providing this new literature, we will make a comparison and add it to the prior work section. We have used the same experimental setting and hyperparameter searching range to implement FAGCN and here are the results we get:
>
> | Test Acc\Models | FAGCN | ACMGCN  | Running Time Per Epoch(ms)\Models | FAGCN| ACMGCN|
> |:-:|:-:|:-:|:-:|:-:|:-:|
> | Cornell     |88.03$\pm$5.6 |  **92.62$\pm$3.04** | Cornell   |8.1ms |8.04ms|
> |Wisconsin | 89.75$\pm$6.37 | **95.37$\pm$2.1** | Wisconsin| 12.9ms | 8.04ms|
> |Texas| 88.85$\pm$4.39 | **95.08$\pm$1.8** |Texas|8.8ms | 8.17ms|
> |Film| 31.59$\pm$1.37 | **41.48$\pm$0.78**| Film| 45.4ms | 9.29ms|
> |Chameleon|  49.47$\pm$2.84|  **67.79$\pm$1.79** |Chameleon |8.4ms| 9.33ms|
> |Squirrel | 42.24$\pm$1.2 | **52.86$\pm$1.96**| Squirrel| 16ms | 12.15ms|
> |Cora |88.85$\pm$1.36  |  **89.11$\pm$0.87**| Cora |8.4ms | 9.16ms|
> |CiteSeer| **82.37$\pm$1.46** | 82.16$\pm$0.84| CiteSeer |9.4ms | 9.48ms|
> |PubMed| 89.98$\pm$0.54 | **90.72$\pm$0.7**| PubMed|14.5ms  |9.54ms|
>
> ACM-GCN ourperform FAGCN except on Citeseer.
>
>
> Q3: "Why there is a little bump in Figure 2d at the high values of H_agg?"
>
> R3: As stated in line 156, it is because of numerical perturbation. We re-implement the experiment by generating 10 graphs per homophily level and the perturbation can be reduced.

---

> ### Author Response · Authors · 2021-09-19
> **Final Confirmation with Reviewer bgkm**
>
> Dear Reviewer bgkm,
>
> Hope you are doing great recently. We are thankful for your positive rating and your 20 hours spending on reading and evaluating our paper. We send this message to confirm if you have more questions or comments on our paper. Please feel free to let us know. Thanks.
>
> Best,
>
> Authors

---

### Official Review · Reviewer_v2mj · 2021-07-16

**Rating:** 5
**Confidence:** 4

**Summary:**

In this paper, the authors investigate how heterophily affects the performance of GNNs. Specifically, they first show that not all cases of heterophily are harmful. Then, they propose some new metrics to better characterize the graph properties and their relations to GNNs’ performance. Finally, the authors propose a framework named Adaptive Channel Mixing to address some kinds of harmful heterophily. This paper provides some new perspectives on understanding and addressing heterophily.

**Main Review:**

Strong points
+ This paper provides new perspectives on the relation between heterophily and the performance of GNNs.
+ A new method named Adaptive Channel Mixing is proposed, which is demonstrated to address some kinds of harmful heterophily.

Weak points:
-	Overall, the connection between the investigation part (heterophily; new metrics) and the methodology part (new model) seems not strong. It would be better if the authors could describe their connection more clearly.
-	In section 3.2, it would be better if the authors could provide more details of the data generation process. Furthermore, it would be helpful if the authors could provide some explanations of the U-shape of these curves.

-	 In section 4, the authors claim that “diversification operation to address harmful heterophily”. However, it is not clear what is “harmful heterophily”. It would be helpful if the authors could provide a more detailed characterization of “harmful heterophily”. The case stays true for the proposed ACM framework. What kinds of “harmful heterophily” can it address? How does ACM address it?

-	It is not clear what Theorem 2 indicates. It would be better if the authors could provide more explanation about this theorem and what it demonstrates.


**Time Spent Reviewing:**

2.5

---

> ### Author Response · Authors · 2021-08-10
> **Response to Reviewer v2mj**
>
> Thanks for your valuable comments and here are our responses to your concerns:
>
> Q1: "Overall, the connection between the investigation part (heterophily; new metrics) and the methodology part (new model) seems not strong. It would be better if the authors could describe their connection more clearly."
>
>
> R1: With the proposed similarity matrix in investigation part, we figure out why high-pass filter is important to address some cases of heterophily. From on the analysis, we argue that low-pass filter and high-pass filter should be used together for feature extraction, which lead us to the filterbank method. We generalize filterbank method and propose ACM framework which includes low-pass, high-pass and identity channels.
>
>
> Q2: "(1) In section 3.2, it would be better if the authors could provide more details of the data generation process. (2) Furthermore, it would be helpful if the authors could provide some explanations of the U-shape of these curves."
>
>
> R2:
> (1)
> Specifically in our synthetic experiments, for a given $h$, we generate node degree $d_v$ for nodes in each class from multinomial distribution with $\texttt{numpy.random.multinomial(800/h, numpy.ones(400)/400, size=1)[0]}$. For a sampled $d_v$, we generate intra-class edges from $\texttt{numpy.random.multinomial(h$d_v$, numpy.ones(399)/399, size=1)[0]}$ (does not include self-loop) and inter-class edges from $\texttt{numpy.random.multinomial((1-h)$d_v$, numpy.ones(1600)/1600, size=1)[0]}$. We will release the code and generated data later.
>
> (2) Just like the example of bipartite graph shown in Figure 1, GNNs do not necessarily perform bad on graphs with low edge homophily values. And the U-shaped curves verifies our claim that when edge homophily value is low, the nodes in different classes can still be highly distinguishable like Figure 1.
>
> Q3: "...(1) it is not clear what is “harmful heterophily”. It would be helpful if the authors could provide a more detailed characterization of “harmful heterophily”. (2) The case stays true for the proposed ACM framework. What kinds of “harmful heterophily” can it address? How does ACM address it?"
>
>
> R3:
> (1) Generally, harmful heterophily means the heterophily structure that makes a graph-aware model underperform its corresponding graph-agnostic model.
>
> (2) The parts of harmful heterophily that can be addressed by high-pass filter are characterized by the diversification distinguishable nodes as stated in definition 2. And the proportion of nodes that can potentially gain benefits from the high-pass channel is indicated by the diversification distinguishability value. Actually, in ACM framework, even when the additional high-pass and identity channels are not beneficial, the nodewise channel mixing mechanism can help us to learn a model that is not worse than the vanilla GNN model which only has a single low-pass channel.
>
>
> Q4: "It is not clear what Theorem 2 indicates. It would be better if the authors could provide more explanation about this theorem and what it demonstrates."
>
>
> R4: It theoretically demonstrates the importance of high-pass filter,
> which leads to the filterbank method and ACM framework.

---

> ### Author Response · Authors · 2021-09-19
> **Final Confirmation with Reviewer v2mj**
>
> Dear Reviewer v2mj,
>
> Hope everthing is fine with you. We are grateful for your valuable comments and constructive suggestions. We would like to make a final confirmation with you to see if our response is satisfactory and whether you have any remaining concern on our paper or not. Please do not hesitate to let us know. Thanks.
>
> Best,
>
> Authors

---

### Official Review · Reviewer_H6yV · 2021-07-17

**Rating:** 6
**Confidence:** 3

**Summary:**

Graph homophily and heterophily recently gain a lot of attention from the community. First, this paper proposes an interesting metric, which considers the general structure of graph convolutional networks, to measure the potential performance of GCNs on a certain type of graph. The metric shows a strong correlation with the accuracy of graph convolutional networks. They further analyze the characteristics of nodes where the GCN may not work properly that also match the synthetic results. Based on these observations, the adaptive channel mixing framework is proposed. The proposed framework utilizes the high/low-frequency filterbanks and shows strong performance on real-world datasets.


**Limitations And Societal Impact:**

Please check the main review.


**Main Review:**

I liked reading this paper. The paper is easy to follow, and the analysis is intuitive and non-trivial, although the configuration is relatively simple. The proposed method with the high/low-frequency filter bank is not new, but the theoretical analysis makes the framework convincing. Some questions and limitations are addressed below:

- What is the correlation between the proposed metric and test accuracy with the 11 real-world datasets? Does it match the observation from synthetic datasets?
- The main analysis is based on the random walk Laplacian, which is not widely used than the symmetric Laplacian. Will it be impossible to draw a similar definition and theorem from the symmetric one?
- Although Theorem 1 shows the situation where the message passing architecture may fail to achieve a good performance, the analysis does not take into account the existence of a learnable weight matrix. What would be the role of the weight matrix, and why can it not overcome these situations?


**Time Spent Reviewing:**

5

---

> ### Author Response · Authors · 2021-08-10
> **Response to Reviewer H6yV**
>
> Thanks for your positive comments and here are our responses to your concerns:
>
> Q1:" What is the correlation between the proposed metric and test accuracy with the 11 real-world datasets? Does it match the observation from synthetic datasets?"
>
>
> R1:
> There are three key factors that influence the test accuracy of GNNs in real world tasks: labels, features and graph structure. The (modified) aggregation homophily tries to investigate how the graph structure will influence the test accuracy with fixed labels and features. And their correlation is verified through the synthetic experiments.
>
> In real-world tasks, we need to consider features and feature-graph consistency as well. With aggregation similarity score of the features $S_\text{agg}\left(S(I,X)\right)$ and aggregated features $S_\text{agg}\left(S(\hat{A},X)\right)$ listed in appendix G, our methods open up a new perspective on analyzing and comparing the performance of graph-aware models and graph-agnostic models in real-world tasks. Here are 2 examples.
>
> Example 1: GCN (graph-aware model) underperforms MLP-2 (graph-agnostic model) on $\textit{Cornell, Wisconsin, Texas, Film}$. Based on the aggregation homophily, the graph structure is not the main cause of the performance degradation. And from Table 6 in Appendix G, we can see that the $S_\text{agg}\left(S(\hat{A},X)\right)$ for the above 4 datasets are lower than their corresponding $S_\text{agg}\left(S(I,X)\right)$, which implies that it is the feature-graph inconsistency that causes the performance degradation.
>
> Example 2: According to the widely used analysis method based on node or edge homophily, the graph structure of $\textit{Chameleon}$, and $\textit{Squirrel}$ are heterophilous and bad for GNNs. But in practice, GCN outperforms MLP-2 on those 2 datasets. Traditional homophily metrics fail to explain such phenomenon but our method can give an explanation from different angles: For Chameleon, its modified aggregation homophily is not low and its $S_\text{agg}\left(S(\hat{A},X)\right)$ is higher than its $S_\text{agg}\left(S(I,X)\right)$ which means its label-graph consistency and feature-graph consistency help the graph-aware model obtain the performance gain; for Squirrel, its modified aggregation homophily is low but its $S_\text{agg}\left(S(\hat{A},X)\right)$ is higher than its $S_\text{agg}\left(S(I,X)\right)$ which means although its label-graph consistency is bad but the feature-graph consistency is the key factor to help the graph-aware model perform better.
>
> We also need to point out that (modified) aggregation similarity score, $S_\text{agg}\left(S(\hat{A},X)\right)$ and $S_\text{agg}\left(S(I,X)\right)$ are not deciding or threshold values because they do not consider the non-linearity structure in the features. In practice, a low score does not tell us the GNN models will definitely perform bad.
>
> Q2:"...The main analysis is based on the random walk Laplacian, which is not widely used than the symmetric Laplacian. Will it be impossible to draw a similar definition and theorem from the symmetric one?"
>
>
> R2: The definitions of the similarity matrix, (modified) aggregation similarity score and diversification distinguishability value can be extended to symmetric normalized Laplacian or other aggregation operations. It is possible to extend Theorem 1 to symmetric normalized Laplacian, while it is unlikely to extend Theorem 2, because we need a condition that the row sum of the Laplacian is not greater than 1 based on the proof in Appendix D. This condition is guaranteed for random walk normalized Laplacian but not for symmetric normalized Laplacian.
>
>
> Q3: "Although Theorem 1 shows the situation where the message passing architecture may fail to achieve a good performance, the analysis does not take into account the existence of a learnable weight matrix. What would be the role of the weight matrix, and why can it not overcome these situations?"
>
>
> R3：We derive the similarity matrix by considering the update direction of the learnable weight matrix. By setting $X=Z$, for a given perfect set of features (an ideal condition), the aggregation homophily actually measures the proportion of node that will move to the correct direction, i.e. putting higher weights to the nodes in the same class, after one backpropagation step over the learnable weight matrix.

---

> > ### Comment · Reviewer_H6yV · 2021-09-05
> > **I'll keep my rating**
> >
> > I have read all the other reviews and the responses from the authors and decided to keep my rating. The authors seem to address the questions raised by the reviewers properly. Although there are some minor concerns about the representation of the paper, I believe it will be updated properly based on the reviews.

---

> > > ### Author Response · Authors · 2021-09-05
> > > **Thanks**
> > >
> > > Thanks for your positive comments and we will keep revising our paper based on the feedback from all reviewers.
> > >
> > > Best,
> > >
> > > Authors

---

### Official Review · Reviewer_7tx9 · 2021-07-21

**Rating:** 5
**Confidence:** 5

**Summary:**

The paper makes two specific points:
1. First the current reliance on homophily scores to distinguish which graphs are likely to perform well for a particular task is not precise. The paper proposes an alternative metric which seems to correlate better with performance.
2. The paper takes inspiration from their proposed metric to define filter banks that separate high pass and low pass information separately and mixes them for the final desired task.

**Limitations And Societal Impact:**

The authors discuss the limitation of the proposed Channel Mixing model through the limitations of Diversification Distinguishability metric. However, as said earlier, the connection between Diversification Distinguishability and Channel Mixing model is not very clear and hence it is unclear how the said limitations. For example, they calculate DD_{A,X}(G) for Pubmed to be 0.1889 which is lower DD_{A,X}(G) for other datasets (Appendix G), but in the experimental section, we see that ACM-GCN performs better than other models on Pubmed. This discrepancy in results is not very clear.

**Main Review:**

First of all, I would like to state the paper is well written and easy to understand, however, there are some gaps in the write up that makes it difficult to logically parse the whole paper.

1. The proof for Theorem 1 is not very detailed and misses steps that are not very clear to me. For example, when computing E[\hat{A}Z], it seems to me that the authors are assuming E[X/Y] = E[X]/E[Y] which is not true, or they are using taylor approximation in which case, the approximation holds only if Cov(X, Y) ~ 0 and Var(Y) ~ 0, both of which I am not sure are true. It would be good for the authors to elaborate on the steps involved to verify the correctness of the stated theorems. (Not only that step, but also how E[S(\hat{A}, Z)] is computed as well, In general, it would be good if all proofs are given in sufficient detail)
2. The authors define Diversification Distinguishability based on their proposed metric. However, later on they state that it is the same as low and high frequency information from graph signals. This part is not very clear at all.
3. In experiments, the authors have used datasets from Geom-GCN paper but it is known that the authors of Geom-GCN misquoted the train/val/test split in their paper, it is not 60/20/20, but rather 48/32/20. So, did the authors use the 60/20/20 split in their experiment or the 48/32/20 split?
4. The GPR-GNN model also attempts combine various frequencies in an amiable fashion for the task. It is unclear as to why this method achieves higher performance than GPR-GNN model. In fact, there is another work which attempts to do something similar [A]. However, this method is compared with in the paper. It would be good to have some reasoning for why models like GPR-GNN fall behind the proposed approach.

[A] Beyond Low-frequency Information in Graph Convolutional Networks, Deyu Bo, Xiao Wang, Chuan Shi, Huawei Shen, https://arxiv.org/abs/2101.00797

**Time Spent Reviewing:**

9

---

> ### Author Response · Authors · 2021-08-10
> **Response to Reviewer 7tx9**
>
> Thanks for your valuable comments and here are our responses to your concerns.
>
> Q1: "...when computing E[\hat{A}Z], it seems to me that the authors are assuming E[X/Y] = E[X]/E[Y] which is not true, or they are using taylor approximation in which case,
> the approximation holds only if Cov(X, Y) ~ 0 and Var(Y) ~ 0, both of which I am not sure are true..."
>
> R1: We do not use the assumption "E[X/Y] = E[X]/E[Y]" or Taylor approximation. The confusion might come from the description of data generation process in synthetic experiments and Theorem 1 and here is a more detailed description:
> 1. For each node $v$, we first randomly generate its degree $d_v$ which satisfies $\mathbb{E}\left[ d_v \right] = d$.
> 2. Given $d_v$, we sample $hd_v$ intra-class edges and $(1-h)d_v$ inter-class edges.
> Then, we have $\mathbb{E}\left[(\hat{A}Z)_{v,c}\right] =  \mathbb{E}\left[\sum\limits_\{k \in \mathcal{V}\} \hat{A}_\{v,k\} \textbf{1}_\{\\{Z_\{k,:\}= e_c^T\\}\}\right] = \sum\limits_\{k \in \mathcal{V}\} \mathbb{E}\left[ \frac{1}{d_v+1} \textbf{1}_\{\\{Z_\{k,:\}= e_c^T\\}\}\right]$.
>
> When $v$ is in class  c, we have $\mathbb{E}\left[ \frac{hd_v+1}{d_v+1} \right] = \mathbb{E}\left[ \frac{h(d_v+1)+1-h}{d_v+1} \right] =  \mathbb{E}\left[h+ \frac{1-h}{d_v+1} \right] = \frac{hd+1}{d+1}$.
>
> When $v$ is not in class  c, we have $\mathbb{E}\left[ \frac{(1-h)d_v}{(C-1)(d_v+1)} \right] = \mathbb{E}\left[ \frac{1-h}{C-1}- \frac{(1-h)}{(C-1)(d_v+1)} \right] = \frac{(1-h)d}{(C-1)(d+1)}$.
>
> More specifically in our synthetic experiments, for a given $h$, we generate node degree $d_v$ for nodes in each class from multinomial distribution with $\texttt{numpy.random.multinomial(800/h, numpy.ones(400)/400, size=1)[0]}$. For a sampled $d_v$, we generate intra-class edges from $\texttt{numpy.random.multinomial(h$d_v$, numpy.ones(399)/399, size=1)[0]}$ (does not include self-loop) and inter-class edges from $\texttt{numpy.random.multinomial((1-h)$d_v$, numpy.ones(1600)/1600, size=1)[0]}$. We will release the code and the generated data later.
>
> Q2: "The authors define Diversification Distinguishability based on their proposed metric. However, later on they state that it is the same as low and high frequency information from graph signals. This part is not very clear at all."
>
> R2：Node-level Aggregation operation is equivalent to applying a low-pass filter to capture the low frequency information of the signal and node-level diversification operation is equivalent to applying a high-pass filter to extract high frequency information of graph signal. The main purpose of lines 196-198 is to connect sections 4.1 and 4.2 and to lead out the concept of filterbank, but we find that this sentence might cause some unnecessary misunderstanding. We will refine it and move it to section 4.2 to avoid confusion.
>
> Q3:"... So, did the authors use the 60/20/20 split in their experiment or the 48/32/20 split?..."
>
> R3: We use the datasets provided by Geom-GCN but the random splits are 60/20/20 as provided by GPRGNN. The reason is that we find random splits and fixed splits might cause different results and fixed splits might lead to some unnecessary overfitting. Thus, we choose the splitting method from a recent published paper, GPRGNN; and to make a fair comparison, we use the 60/20/20 ramdom splitting method to run all the experiments.
>
> Q4: "...In fact, there is another work which attempts to do something similar [A]. However, this method is compared with in the paper. It would be good to have some reasoning for why models like GPR-GNN fall behind the proposed approach."
>
> R4: Thanks for providing this new literature [A] and we will add it to the prior work section.
>
> Comparison with GPRGNN: GRPGNN is $\mathbf{Z}=\sum_\{k=0\}^{K} \gamma_\{k\} \mathbf{H}^{(k)}$, where $\gamma_\{k\} \in \mathbb{R}$ is a trainable scaler parameter. ACM differs from GPRGNN mainly in having a nodewise channel mixing mechanism but GPRGNN only has one scalar parameter to control the combination weight for information from each hop. When the scalar parameter becomes negative, it applies to all nodes. But from our analysis, different nodes have different demands for the information from different channels. Thus, we design the nodewise channel mixing mechanism.
>
> Comparison with FAGCN [A]: FAGCN defines a learnable pairwise aggregation weight $\alpha_\{ij\}^G \in [-1,1]$, which is similar to GAT but allows negative weights. ACM first does aggregation and diversification operation without learning any weights, and then learns nodewise channel mixing weights to select from different channels. Moreover, ACM is derived from filterbank method and this allows us to find out a special identity channel which is empirically effective. FAGCN is inspired by an experimental investigation which shows that both low-frequency and high-frequency signals are important.
>
> Q5: "...(1) the connection between Diversification Distinguishability and Channel Mixing model is not very clear and hence it is unclear how the said limitations. (2) For example, they calculate DD_{A,X}(G) for Pubmed to be 0.1889 which is lower DD_{A,X}(G) for other datasets (Appendix G), but in the experimental section, we see that ACM-GCN performs better than other models on Pubmed. This discrepancy in results is not very clear."
>
> R5:
> (1)
> Diversification Distinguishability gives us a reference value which indicates the proportion of node that can potentially be helped by applying high-pass filter. Different channels are useful for different nodes: for some nodes, low-pass and high-pass channels are both helpful and in other cases one of the low- and high-pass channel is helpful.  Since we do not know which node needs which channel, we need a nodewise channel mixing mechanism to learn which channel we should focus more.
>
> Besides the above situations, we can also have some cases that neither low-pass nor high-pass channel can help with distinguishing nodes as discussed in lines 231-236 and Appendix F. This means there exist some cases that ACM framework will not work well no matter which channel is focused on. This is the limitation of ACM framework and we leave this challenge to future work.
>
> (2) Graph Diversification Distinguishability value measures the proportion of nodes that can potentially get benefits from using high-pass filter. Thus, even with $DD_\{\hat{A},Z\}(G) = 0.1889$, this does not mean GNN models will not get performance improvement from high-pass filter. In addition, due to the channel mixing mechanism, even the nodes do not gain benefits from high-pass channel, they can learn to extract useful information from low pass and identity channels. Actually, we need to clarify that $DD_\{\hat{A},Z\}(G)$ is a reference value instead of a deciding value.

---

> > ### Author Response · Authors · 2021-08-18
> > **Performance Comparison on Fixed 48/32/20 Splits**
> >
> > To further address your concern on the performance of ACM-GNNs under the fixed 48/32/20 splits provided by GeomGCN, we evaluate ACN-GCN, ACM-SGC-1 and ACM-SGC-2 under the same setting as GeomGCN and make comparison with some state-of-the-art models:
> >
> > | Test Acc($\pm$Std)\Models | GeomGCN | H$_2$GCN | GPRGCN | FAGCN | ACM-SGC-1 | ACM-SGC-2 |ACM-GCN |
> > |:-:|:-:|:-:|:-:|:-:|:-:|:-:|:-:|
> > |Cornell|60.54 $\pm$ 3.67 |82.70 $\pm$ 5.28 |78.11 $\pm$ 6.55 |76.76 pm 5.87 |82.43 $\pm$ 5.44 |82.43 $\pm$ 5.44  | **85.95 $\pm$ 5.64** |
> > |Wisconsin| 64.51 $\pm$ 3.66| 87.65 $\pm$ 4.98 |82.55 $\pm$ 6.23 |79.61 pm 1.58 |86.47 $\pm$ 3.77|86.47 $\pm$ 3.77|**88.43 $\pm$ 3.22**|
> > |Texas|66.76 $\pm$ 2.72 |84.86 $\pm$ 7.23 |81.35 $\pm$ 5.32 |76.49 pm 2.87|81.89 $\pm$ 4.53 |81.89 $\pm$ 4.53 |  **87.84 $\pm$ 4.4**|
> > |Film|31.59 $\pm$ 1.15 |35.70 $\pm$ 1.00 |35.16 $\pm$ 0.9 |34.82 pm 1.35 |35.49 $\pm$ 1.06 |36.04 $\pm$ 0.83   |  **36.28 $\pm$ 1.09** |
> > |Chameleon|60.00 $\pm$ 2.81 |60.11 $\pm$ 2.15 | 62.59 $\pm$ 2.04 |46.07 pm 2.11 |63.99 $\pm$ 1.66 |59.21 $\pm$ 2.22  |**66.93 $\pm$ 1.85**|
> > |Squirrel|38.15 $\pm$ 0.92 |36.48 $\pm$ 1.86| 46.31 $\pm$ 2.46 |30.83 pm 0.69 |45.00 $\pm$ 1.4 |40.02 $\pm$ 0.96 | **54.4 $\pm$ 1.88**|
> > |Cora|85.35 $\pm$ 1.57 |87.87 $\pm$ 1.20 | 87.95 $\pm$ 1.18 |**88.05 pm 1.57** |86.9 $\pm$ 1.38|87.69 $\pm$ 1.07   | 88.01 $\pm$ 1.08|
> > |CiteSeer|**78.02 $\pm$ 1.15** |77.11 $\pm$ 1.57 | 77.13 $\pm$ 1.67 |77.07 pm 2.05 |76.73 $\pm$ 1.59 |76.59 $\pm$ 1.69 |77.32 $\pm$ 1.7 |
> > |PubMed|89.95 $\pm$ 0.47 |89.49 $\pm$ 0.38 | 87.54 $\pm$ 0.38|88.09 pm 1.38 |88.49 $\pm$ 0.51 | 89.01 $\pm$ 0.6  | **90.00 $\pm$ 0.52**|
> > |Average Rank|5.33|3.33|4.22|5.56 |3.89 |4.11|**1.22**|
> >
> > We can see that ACM-GCN achieves the best performance, especially on the graphs with heterophily. ACM-SGC-1 and ACM-SGC-2 match or exceed the state-of-the-art models as well. In general, the baseline GNN models can get permormance boost under ACM framework on solving node classification tasks, particularly for graphs with heterophily.
> >
> > If more information is needed to address your concerns, please let us know. Thanks.

---

> ### Comment · Reviewer_7tx9 · 2021-09-02
> **Response to rebuttal**
>
> I thank the authors for giving a detailed response to all the concerns raised in the review.
>
> R1: In the response, the authors again seem to have used E[1/X] = 1/E[X] (which is not true). To make it more clear, the term d_v disappears from the denominator and d appears in its place. That step looks wrong.
>
> R2: "Node-level Aggregation operation is equivalent to applying a low-pass filter to capture the low frequency information of the signal and node-level diversification operation is equivalent to applying a high-pass filter to extract high frequency information of graph signal."
> Sorry, but it is still not clear how the defined Diversification Distinguishability part is connected to the node aggregation and diversification operation, which thereby leads to the low-pass and high-pass filtering.
>
> R4: It is good to see comparison with other baseline methods and the proposed method.
>
> A] While the differences between ACM and other methods are clear, it is not made clear as to what are the gaps in these approaches that are getting fixed with ACM. For example, why is node-wise channel mixing important? GPR-GNN due to utilizing polynomial filter is capable of expressing any latent filter characteristics, what is the advantage that node-wise channel mixing gives that GPRGNN lacks? Why is node-wise channel mixing working better than pairwise aggregation weights in FAGCN?
>
> B] I looked at the revised table, and I find that the GPRGNN numbers look far lower than what is reported in the GPRGNN paper, particularly for Chameleon and Squirrel.
>
> Finally, I again thank the authors for giving such a detailed response and doing additional experiments as needed. However, it might be worthwhile for the authors to look at the related approaches more carefully, and position their paper more accurate in the context of these works. In light of this, I will retain my current evaluation and encourage the authors to resubmit their work clarifying the steps in proofs to their theorems and contextualizing the work with respect to works like FAGCN, GPRGNN etc.

---

> > ### Author Response · Authors · 2021-09-02
> > **Response2 to Reviewer 7tx9**
> >
> > Thanks for your careful evaluation and valuable suggestion. Here are our responses to your remaining concerns.
> >
> >
> > **Q1:** In the response, the authors again seem to have used E[1/X] = 1/E[X] (which is not true). To make it more clear, the term d_v disappears from the denominator and d appears in its place. That step looks wrong.
> >
> > **R1:** Thanks for pointing this out. We would like to first provide a revised description of the assumption to avoid confusion and explain the motivation of theorem 1.
> >
> > A]: A revised version of the assumption in theorem 1 "Suppose there are $C$ classes in the graph $\cal G$ and **node degrees are $d$ for all nodes. Given $d$, edges for each node are i.i.d. generated**, such that each edge of any node has probability $h$ to connect with nodes in the same class and probability $1-h$ to connect with nodes in different classes. Let the aggregation operator $\hat{A} = \hat{A}_\text{rw}$. Then, ..."
> >
> > We also revise the setting of the synthetic experiments to "...we have 5 classes with 400 nodes in each class. **For each node, we randomly generate 2 intra-class edges and [$\frac{2}{h} -2$] inter-class edges**". For the updated figures of the synthetic experiments, please check this anonymous repo: https://github.com/AnonymousAuthors2021/NeurIPS2021_Rebuttal
> >
> > B]: We try to align the assumption of Theorem 1 with the data generation process in the synthetic experiments and the goal of the synthetic experiments together with theorem 1 is just to verify the advantage of the (modified) aggregation homophily over the existing metrics. The contribution of this part is focused on the (modified) aggregation homophily. Thus, slightly revising the assumption and data generation process with stronger conditions does not change the function of the synthetic experiments and theorem 1.
> >
> > **Q2:** "Node-level Aggregation operation is equivalent to applying a low-pass filter to capture the low frequency information of the signal and node-level diversification operation is equivalent to applying a high-pass filter to extract high frequency information of graph signal." Sorry, but it is still not clear how the defined  Diversification Distinguishability part is connected to the node aggregation and diversification operation, which thereby leads to the low-pass and high-pass filtering.
> >
> > **R2:** Diversification Distinguishability is a measure of how much the **additional diversification (high-pass) channel** can potentially be helpful based on the proposed similarity matrix (Diversification Distinguishability has nothing to do with aggregation channel). Together with the original aggregation (low-pass) channel used in traditional methods, we get a filterbank. Inspired by the filterbank method, we design ACM framework.
> >
> > **Q3:**
> >
> >  A] "...why is node-wise channel mixing important? GPR-GNN due to utilizing polynomial filter is capable of expressing any latent filter characteristics, what is the advantage that node-wise channel mixing gives that GPRGNN lacks? Why is node-wise channel mixing working better than pairwise aggregation weights in FAGCN?"
> >
> > B] "I looked at the revised table, and I find that the GPRGNN numbers look far lower than what is reported in the GPRGNN paper, particularly for Chameleon and Squirrel."
> >
> > **R3:**
> >
> > A]: As shown in equation 13, the node-wise channel mixing is equivalent to learning a parametric diagonal matrix which is more expressive than the scalar parameters used in GPRGNN.
> > And from our analysis, we find that different nodes have different needs for the information from different channels. For example, in figure 3, node 2 needs more aggregation information and less diversification information
> > while 1 and 3 need more diversification information and less aggregation information. In GPRGNN, once the learnable scalar becomes positive or negative, it is shared by all nodes and thus, GPRGNN will not work well in this example.
> >
> >
> > As for FAGCN, in some sense, learning pairwise aggregation weights is equivalent to learning two new low-pass filters and then do a subtraction of these two filters. Again this is similar to GPRGNN and does not contain a node-wise channel mixing mechanism.
> >
> > B]: We would like to clarify that the results we provided in our rebuttal is the performance on the **fixed 48/32/20 splits** provided by Geom-GCN. The results of GPRGNN in its original paper are tested on **random 60/20/20 splits**. For the experiments In our paper, we use the open source splitting code provided by the authors of GPRGNN, please see: https://github.com/jianhao2016/GPRGNN/blob/f4aaad6ca28c83d3121338a4c4fe5d162edfa9a2/src/utils.py#L16
> >
> > The performance of GPRGNN on fixed 48/32/20 splits is not as good as its performance on random 60/20/20 splits.
> >
> > Please do not hesitate to reply if you need more information to address you concerns. Thanks.
> >
> > Best,
> >
> > Authors

---

> > > ### Author Response · Authors · 2021-09-04
> > > **Revised Version of Theorem 1 and Its Proof**
> > >
> > > To further address your concerns on theorem 1 and its proof, we make a revision based on your feedback. Please check this anonymous repo for the updated version:
> > >
> > > https://github.com/AnonymousAuthors2021/NeurIPS2021_Rebuttal#revised-theorem-1-and-its-proof
> > >
> > > Please let us know if you have remaining questions for theorem 1 or other parts in our paper. Thanks.
> > >
> > > Best,
> > >
> > > Authors

---

> ### Comment · Reviewer_7tx9 · 2021-09-05
> **Thanks for the update**
>
> I thank the authors for the quick response to all my queries.
>
> > A revised version of the assumption in theorem 1 "Suppose there are classes in the graph and node degrees are for all nodes.
> > Given , edges for each node are i.i.d. generated, such that each edge of any node has probability to connect with nodes in the
> > same class and probability to connect with nodes in different classes. Let the aggregation operator. Then, ..."
>
> Yes, if you assume a d-regular graph in your Theorems, it does fix things. Although, unsure of what it says about general graphs which are usually not regular. That is fine I suppose.
>
> > As shown in equation 13, the node-wise channel mixing is equivalent to learning a parametric diagonal matrix which is
> > more expressive than the scalar parameters used in GPRGNN. And from our analysis, we find that different nodes
> > have different needs for the information from different channels. For example, in figure 3, node 2 needs more aggregation
> > information and less diversification information while 1 and 3 need more diversification information and less
> > aggregation information. In GPRGNN, once the learnable scalar becomes positive or negative, it is shared by all nodes
> > and thus, GPRGNN will not work well in this example.
>
> I think there are some gaps here. You can find a polynomial filter that gives as good a performance as node-wise mixing model. Hence, node-wise mixing is not necessarily more expressive than a polynomial filter.
>
> > As for FAGCN, in some sense, learning pairwise aggregation weights is equivalent to learning two new low-pass filters and
> > then do a subtraction of these two filters. Again this is similar to GPRGNN and does not contain a node-wise channel mixing
> > mechanism.
>
> As per my understanding of FAGCN, you have filter weights at every edge, while it may not contain node-wise mixing mechanism, it does contain mixing of information at edge level.
>
> > We would like to clarify that the results we provided in our rebuttal is the performance on the fixed 48/32/20 splits provided by
> > Geom-GCN. The results of GPRGNN in its original paper are tested on random 60/20/20 splits. For the experiments In our
> > paper, we use the open source splitting code provided by the authors of GPRGNN, please see:
> > https://github.com/jianhao2016/GPRGNN/blob/f4aaad6ca28c83d3121338a4c4fe5d162edfa9a2/src/utils.py#L16
>
> This paper [A] also works with 48/32/20 split and reports somewhat higher GPRGNN numbers on Chameleon than what is reported in this paper.
>
> [A] Simple Truncated SVD based Model for Node Classification on Heterophilic Graphs, DLG-KDD, https://arxiv.org/abs/2106.12807

---

> > ### Author Response · Authors · 2021-09-05
> > **Response 3 to the Remaining Concerns of Reviewer 7tx9**
> >
> > Thanks for your prompt reply and constructive comments. Here are our responses to your remaining concerns.
> >
> > **Q1**: "I think there are some gaps here. You can find a polynomial filter that gives as good a performance as node-wise mixing model. Hence, node-wise mixing is not necessarily more expressive than a polynomial filter."
> >
> > **R1**: To explain the difference between channel mixing mechanism and the learning mechanism in GPRGNN, we first rewrite GPRGNN as $\mathbf{Z} = \sum_\{k=0\}^{K} \gamma_\{k\} \mathbf{H}^{(k)} = \sum_\{k=0\}^{K} \gamma_\{k\} I \mathbf{H}^{(k)} = \sum_\{k=0\}^{K} diag(\gamma_\{k\}, \gamma_\{k\},\dots,\gamma_\{k\}) \mathbf{H}^{(k)}$. The node-wise channel mixing mechanism in GPRGNN form is $\mathbf{Z} = \sum_\{k=0\}^{K} diag(\gamma_\{k\}^1,\gamma_\{k\}^2,\dots,\gamma_\{k\}^N) \mathbf{H}^{(k)}$, where $N$ is the number of nodes and $\gamma_\{k\}^i, i=1,\dots,N$ are learnable parametric mixing weights.
> >
> > The effectiveness of the channel mixing mechanism is also empirically verified in the ablation study. Please see table 1 in our paper for reference.
> >
> > **Q2**: "As per my understanding of FAGCN, you have filter weights at every edge, while it may not contain node-wise mixing mechanism, it does contain mixing of information at edge level."
> >
> >
> > **R2**: Yes, to our understanding, instead of using a fixed $\hat{A}$, FAGCN learns a new filter $\hat{A}'$ based on $\hat{A}$. And $\hat{A}'$ can be decomposed as $\hat{A}'=\hat{A}_1'-\hat{A}_2'$, where $\hat{A}_1'$ and $-\hat{A}_2'$ represents positive and negative edge (propagation) information respectively.
> >
> > In our paper, we are not discussing the advantages of using the (edge-level) learned filter $\hat{A}'$ over the fixed filter $\hat{A}$, we are comparing the models with and without node-wise channel mixing mechanism. We think that what we both agree is that FAGCN does not contain node-wise channel mixing mechanism and the channel mixing mechanism is effective in practice. As for the comparison of using the fixed filter with node-wise channel mixing and the (edge-level) learned filter without channel mixing, we believe the empirical evidence on real-world tasks is one of the best ways to compare them. Besides the results on fixed 48/32/20 splits, we can additionally provide the comparison on random 60/20/20 splits. Please check the following table.
> >
> > | Test Acc\Models | FAGCN | ACMGCN  | Running Time Per Epoch(ms)\Models | FAGCN| ACMGCN|
> > |:-:|:-:|:-:|:-:|:-:|:-:|
> > | Cornell     |88.03$\pm$5.6 |  **92.62$\pm$3.04** | Cornell   |8.1ms |8.04ms|
> > |Wisconsin | 89.75$\pm$6.37 | **95.37$\pm$2.1** | Wisconsin| 12.9ms | 8.04ms|
> > |Texas| 88.85$\pm$4.39 | **95.08$\pm$1.8** |Texas|8.8ms | 8.17ms|
> > |Film| 31.59$\pm$1.37 | **41.48$\pm$0.78**| Film| 45.4ms | 9.29ms|
> > |Chameleon|  49.47$\pm$2.84|  **67.79$\pm$1.79** |Chameleon |8.4ms| 9.33ms|
> > |Squirrel | 42.24$\pm$1.2 | **52.86$\pm$1.96**| Squirrel| 16ms | 12.15ms|
> > |Cora |88.85$\pm$1.36  |  **89.11$\pm$0.87**| Cora |8.4ms | 9.16ms|
> > |CiteSeer| **82.37$\pm$1.46** | 82.16$\pm$0.84| CiteSeer |9.4ms | 9.48ms|
> > |PubMed| 89.98$\pm$0.54 | **90.72$\pm$0.7**| PubMed|14.5ms  |9.54ms|
> >
> > **Q3**: "This paper [A] also works with 48/32/20 split and reports somewhat higher GPRGNN numbers on Chameleon than what is reported in this paper."
> >
> >
> > **R3**: Thanks for providing [A]. Our results of GPRGNN on fixed 48/32/20 splits refer to [B] (please see table 7 in the appendix B.3 of [B]). As they mention in their paper, "...We set the same
> > hyperparameters that are provided by the paper or the authors’ github repository, and we match their
> > reported results...", we guess it might be the different hyperparameter settings that cause the discrepancy between the reported results.
> >
> > We have modified the results on $\textit{Cornell, Texas, Wisconsin, Film, Squirrel, Chameleon}$ in the previous table with the results in [A] as you suggest, please take a look. We will also update our paper based on the results in [A].
> >
> > Looking forward to your reply.
> >
> > Best,
> >
> > Authors
> >
> > [B] Yan, Y., Hashemi, M., Swersky, K., Yang, Y., & Koutra, D. (2021). Two Sides of the Same Coin: Heterophily and Oversmoothing in Graph Convolutional Neural Networks. arXiv preprint arXiv:2102.06462. https://arxiv.org/pdf/2102.06462.pdf

---

> ### Author Response · Authors · 2021-09-17
> **Final Confirmation with Reviewer 7tx9**
>
> Dear Reviewer 7tx9,
>
> We hope everything is going well for you. We would like to first appreciate your patient evaluation and we really enjoy the discussion with you during the whole rebuttal period. Since the final decision date is approaching and we haven't heard from you for around 2 weeks, we just send this message to confirm if you still have any concern left. Feel free to let us know and look forward to your reply.
>
> Best,
>
> Authors

---

### Decision · Program_Chairs · 2021-09-27

**Decision:**

Reject

**Comment:**

The ways GNNs handle graph data with variations on homophily/heterophily has received a great deal of attention in the recent literature, and the submissions aims to provide new tools (metrics) for studying this question. The paper did not strike any reviewer as a critical addition to the literature, and concerning issues were surfaced regarding a main proof (Thm 1). After several exchanges between the authors and a reviewer, the authors acknowledged that their proof needed repair, and presented a new set of assumptions that revised the theorem statement and proved the revised statement. After a discussion between the AC and SAC, it was agreed that this sort of a revision is beyond the scope of what can reasonably be handled within the NeurIPS response period. Handling it properly requires a full R&R loop akin to a journal review cycle, where the full set of reviewers are re-activated to consider how the change in assumptions change their interpretation of the significance of the result, not to mention checking the new proof. This is simply beyond what the NeurIPS response period can handle. I therefore encourage the authors to resubmit a revised manuscript to a future venue for a full review, as the work was generally well received.